

# Aggregation kinetics of binary systems containing kaolinite and *Pseudomonas putida* induced by different 1:1 electrolytes: specific ion effects

Zhaoxuan Yu, Rui Tian, Dian Liu, Yekun Zhang and Hang Li

Key Laboratory of Soil Multi-Scale Interfacial Process, College of Resource and Environment, Southwest University, Chongqing, China

## ABSTRACT

**Background**. The interactions between colloidal particles in the binary systems or mixture colloids containing clay minerals and bacteria have important influences on formations and stabilities of soil aggregates, transportations of soil water, as well as biological activities of microorganisms. How the interfacial reaction of metal ions affects their interaction therefore becomes an important scientific issue.

**Methods**. Dynamic light scattering studies on the aggregation kinetics of mixture colloids containing kaolinite and *Pseudomonas putida* (*P. putida*) were conducted in this study.

**Results**. Aggregation could be observed between kaolinite and kaolinite, between kaolinite and *P. putida* when *P. putida* content was less than 33.3%. Additionally, aggregation rates decreased with increasing *P. putida* content. The critical coagulation concentrations and activation energies indicated that there were strong specific ion effects on the aggregation of mixture colloids. Most importantly, the activation energy increased sharply with increasing *P. putida* content, which might result from the lower Hamaker constant of *P. putida* compared with that of kaolinite.

**Contributions**. (1) Strong specific ion effects on mixture colloids aggregation of kaolinite-*P. putida* were observed; (2) the aggregation behavior of mixture colloids was determined by the average effects of mixture colloids, rather than the specific component. This finding provides an important methodological guide for further studies on the colloidal aggregation behavior of mixture systems with organic and inorganic materials.

## INTRODUCTION

Colloids in soil can be divided into inorganic colloids represented by minerals as well as organic colloids represented by microorganisms and organic macromolecules (*Xiong, 1983*). The interactions between soil colloidal particles have important influences on the formation and stability of soil aggregates, the transportation of soil water, the biological activities of microorganisms, and so on (*Borgnino, 2013*; *Loosli et al., 2019*).

Corresponding authors
Rui Tian, tr2016@swu.edu.cn
Hang Li, lihangswu@163.com

Dynamic light scattering (DLS) technique has been widely used in the study with respect to aggregation processes of soil mineral and organic colloids (*Artemyeva et al., 2017*; *Derrendinger & Sposito, 2000*; *Nguyen et al., 2013*; *Nguyen et al., 2017*; *Tian et al., 2014*; *Yan, Cheng & Shang, 2019*; *Zhu et al., 2017*). Moreover, parameters with regard to aggregation kinetics including the critical coagulation concentration (CCC) (*Jia et al., 2013*), aggregation activation energy (*Tian et al., 2014*), ionic polarization (*Gao et al., 2014*), and Hamaker constant (*Luo et al., 2018*) can be obtained by DLS measurement based on the determination of the hydrodynamic diameter of aggregates.

Recent studies have indicated that there are strong specific ion or Hofmeister effects on soil colloids aggregation. Based on the DLS study on the aggregation of montmorillonite colloids in the various alkali ion solutions, *Tian et al. (2014)* found that the CCC values for $Li^+$, $Na^+$, $K^+$, $Rb^+$ and $Cs^+$ (nitrate) were respectively 277.2, 132.8, 80.3, 31.7 and 27.2 mmol/L. For the aggregation of humic acid and "mineral-humic acid" mixture colloids in the divalent ion solutions, similar specific ion effects were also observed (*Gao et al., 2012*; *Gao et al., 2015*). It has long been believed that the difference between ionic radius or ionic hydration radius should be the primary origin of specific ion effects (*DeWalt-Kerian et al., 2017*). However, compared with the difference from ionic electrostatic effects, the difference from ionic radius and ionic hydration radius is merely a quatratic term (*Chen, 2001*), which cannot be used to interpret the above-mentioned up-to-10 times differences between CCC values. A recent study indicated that the ionic dispersion forces might be an important reason for the specific ion effects (*Bostrom, Williams & Ninham, 2001*), which could be important especially when the Coulomb/electrostatic effect was weak (i.e., when the electrostatic field was sufficiently shielded because of high electrolyte concentrations) (*Parsons et al., 2011*). However, *Tian et al. (2014)* observed more pronounced specific ion effects at lower electrolyte concentrations, which obviously cannot be explained by the ionic dispersion forces. A series of studies on the effects of ion interfacial reaction on colloidal particle aggregation have been carried out (*Du et al., 2017*; *Gao et al., 2014*; *Hu et al., 2015*; *Li et al., 2015*; *Liu et al., 2014*; *Tian et al., 2015*; *Xu et al., 2015*). They suggested that the strong non-classical polarization, which was caused by the strong electric field formed by the surface charge of clay minerals in water system, could give a rational explanation for specific ion effects on mineral colloids aggregation. These new studies indicated that this non-classical polarization effects could be as strong as the Coulomb force, and were up to $10^4$ times that of the classical polarization (*Liu et al., 2014*). Moreover, this non-classical polarization not only greatly enhanced the adsorption intensity of the ions onto the surface, but also deeply affected the interaction between the colloidal particles (*Du et al., 2017*).

There are a large number of microorganisms in the soil. Although the volume scale of these microorganisms is larger than the scale defined by colloidal particles, their colloidal properties are still obvious since the particle size of the microorganisms would be 1–1,000 nm in at least one direction (*Sumner, 2000*). This might be the reason why the soil scientists always identified microorganisms as microbial colloids (*Peng et al., 2018*; *Pisarcik et al., 2016*). It is worth noting that microbial colloids are neither mineral colloids nor soil humus colloids. Taking the bacterial cell as an example, a large number of functional groups and surface charges, existing on both interior and exterior hydrophilic surfaces, have similarities

to those generating from the general organic macromolecules and even mineral colloidal surfaces; however, there is a lipid bilayer existing between the interior and exterior surfaces of the bacterial cell membrane.

There have been plenty of studies on "mineral-bacteria" interactions so far (*Diao et al., 2014*; *Hong et al., 2014*; *Qu et al., 2018*; *Wu et al., 2012*; *Zhao et al., 2018*; *Zhao et al., 2012*; *Zhao et al., 2014*), but the effects of ionic polarization, especially non-classical ionic polarization, on the "mineral-bacteria" interaction are rarely reported. A large number of studies have shown that addition of natural organic matter (NOM) leads to changes in the aggregation of inorganic colloids (*Ramirez et al., 2019*; *Sun et al., 2018*). However, there are few studies when it comes to microorganisms. A large number of studies have shown that the addition of NOM leads to changes in the aggregation of inorganic colloids. It has been found that the adsorption of bacteria onto the mineral surfaces is mainly affected by factors such as pH, ionic strength (*Wu et al., 2012*), clay type (*Bellou et al., 2015*), bacteria types and its cell surface properties (*Poortinga et al., 2002*), bacteria growth cycle (*Wu et al., 2014*), bacteria/mineral mass ratio (*Yee, Fein & Daughney, 2000*), and so on (*Tsagkari & Sloan, 2018*). Many theories were used to explain these phenomena. The attachment process of microorganisms onto minerals was described by the Langmuir isotherm equation (*Vasiliadou & Chrysikopoulos, 2011*; *Vasiliadou et al., 2011*) and by the Freundlich isotherm equation (*Chrysikopoulos & Syngouna, 2012*). *Jiang et al. (2007)* found that, compared with $Na^+$, $Mg^{2+}$ was more effective in promoting the adsorption of *Pseudomonas putida* (*P. putida*) on minerals, indicating that electrostatic interaction plays an important role in the adsorption process. *Zhao et al. (2012)* demonstrated that the Derjaguin–Landau–Verwey–Overbeek (DLVO) theory can be used to explain the adsorption of clay minerals (i.e., montmorillonite and kaolinite) by *Escherichia. coli* and *Streptococcus. suis* under very low electrolyte conditions. The adsorption of *Bacillus subtilis* on the phyllosilicates surface could be well explained by the extended DLVO theory (*Hong et al., 2014*). *Parikh & Chorover (2006)* found that P-OFe covalent bonds can be formed between the carboxyl or phosphate groups on the bacteria (*P. putida*) surface and the iron atoms on the surface of hematite and goethite. Meanwhile, the adsorption thermodynamic principles have been widely used in the study on "mineral-bacteria" interactions (*Chen, Rockhold & Strevett, 2003*; *Chen & Zhu, 2005*; *Hendricks, Post & Khairnar, 1979*).

The above analysis indicates that it would be a valuable scientific question to clarify the role of non-classical ionic polarization on the "mineral-bacteria" interaction. In this work, the aggregation behaviors of "kaolinite-*P. putida*" induced by $LiNO_3$, $KNO_3$ and $CsNO_3$ were studied by DLS technique. Whether the aggregation of "mineral-bacteria" mixture colloids would be affected by the specific ion effects was identified firstly, and then how the non-classical ionic polarization effects affect the "mineral-bacteria" interaction was clarified.

## MATERIALS AND METHODS

### Bacterial suspension

*P. putida* (Gram-negative), purchased from China Center for Type Culture Collection, was selected as the test strains. The strain *P. putida* was cultured at 30 °C on Luria-Bertani (LB) liquid medium. The LB medium per liter (pH = 7) contained 10 g Tryptone, 5 g Yeast and 10 g NaCl. *P. putida* was stored in a glycerol tube (30%) at −80 °C, and activated in LB medium at 30 °C and 120 r/min for 7 h until an optical density at 600 nm ($OD_{600}$) of 0.5 was reached before the experiment. one mL of the activation solution was incubated in 250 mL LB medium at 30 °C, 120 r/min for 14 h when the cell reached the stationary phase ($OD_{600} = 2.0$). Then the suspension was freeze-dried (Dong, Zhou 2019). During the aggregation experiments, 10 mg of freeze-dried *P. putida* powders were identically dispersed in the pH = 8.0 sterile water (pre-adjusted by 10 mmol/L NaOH). The bacterial suspension was ultrasonically dispersed with a KQ5200DE ultrasonic disruptor at 40 kHz for 15 min. The hydrodynamic diameter of *P. putida* was measured as $1600 \pm 100$ nm through DLS measurement before adding electrolyte.

### Kaolinite colloidal suspension

Kaolinite (>99.99% pure, purchased from Xuzhou, Jiangsu Province, China) was used in this study. The cation exchange capacity (CEC) and specific surface area were determined to be 37.5 mequiv./kg and 57 $m^2$/g, respectively. The kaolinite colloidal suspension were prepared according to the following procedure (*Xiong, 1983*). 50 g kaolinite particles and 10 mL 100 mmol/L KOH solutions were successively added into a 500 mL beaker, and then diluted with sterile water to 500 mL. After 15 min of intensive sonication, the suspension was further diluted to 5 L using sterile water. The kaolinite colloidal particles with the effective hydrodynamic diameter of less than 300 nm were extracted and collected using the static sedimentation method. The particle density was estimated to be 7.35 g/L by the oven drying method. Then the prepared kaolinite colloidal suspensions were diluted 10 times and the pH value was measured to be $8.0 \pm 0.1$. The hydrodynamic diameter of kaolinite was measured to be $320 \pm 20$ nm.

### Experimental conditions

The experiment was carried out at a temperature of 25 °C and pH 8.0, which can keep the dispersion of both the bacteria and the kaolinite colloids as well as the cell integrity of the bacteria. The selected electrolytes and their concentrations were $LiNO_3$ (0–900 mmol/L), $KNO_3$ (0–700 mmol/L), and $CsNO_3$ (0–450 mmol/L), respectively. The total mass concentration of the colloidal suspension were 300 mg/L, and the mass concentrations of the bacteria in the mixed suspension were set as 0, 10, 20, 50, and 100 mg/L, respectively, indicating the *P. putida* contents were 0%, 3.33%, 6.67%, 16.67%, and 33.33%, respectively.

### Dynamic light scattering measurement

A BI-200SM multi-angle laser light scattering instrument (Brookhaven Instruments Corporation, New York, USA) with a BI-9000AT auto-correlator (Brookhaven Instruments Corporation) was used for the measurement of the aggregates' particle size (e.g., effective

hydrodynamic diameter) which will change during the aggregation process. The power of the laser device equals 15 mW and the laser is vertically polarized with a wavelength of 532 nm. Experimentally, ultrapure water, kaolinite suspensions, bacteria suspensions and the electrolyte ($LiNO_3$, $KNO_3$ or $CsNO_3$) solutions of different concentrations were mixed in the scattering bottle with the total volume of 10 mL. The procedure of aggregation kinetic measurement was that: each colloidal (including single and mixed colloids) suspension was prepared with a 2 min sonication before adding the electrolyte. After adding electrolyte, the information about particles size and the size distribution was recorded automatically by DLS measurement. The duration for each experiment was 30 min because the light scattering intensity gradually became unstable after 30 min.

CCC, the minimum electrolyte concentration required for the Diffusion-Limited colloid aggregation (DLCA) regime (*Lin et al., 1989*), is an important parameter for characterizing the colloidal aggregation. The CCC value directly reflects the difference in the colloidal stability. Based on the research of *Jia et al. (2013)*, the aggregation kinetics of colloidal particles in electrolyte solutions can be described by the total average aggregation (TAA) rate, which was expressed as,

$$\tilde{v}_T(f_0) = \frac{1}{t_0} \int_0^{t_0} \frac{D(t) - D_0}{t} dt \qquad (1)$$

where $\tilde{v}_T(f_0)$ (nm/min) is the TAA rate from $t = 0$ to a given time $t = t_0$ ($t_0 > 0$) which is committed to a time limit of the aggregation process; $f_0$ (mmol/L) is the electrolyte concentration; $D_0$ and $D_{(t)}$ (nm) are respectively the effective hydrodynamic diameters of aggregates at the beginning and at time $t_0$.

*Tian et al. (2014)* showed that the activation energy ($\Delta E$) and the TAA rate ($\tilde{v}_T(f_0)$) are correlated with the expressions as

$$\Delta E = -RT \ln \frac{\tilde{v}_T(f_0)}{\tilde{v}_T(CCC)} \qquad (2)$$

where, $R$ (J/mol K) is the gas constant and $T$ (K) is the absolute temperature.

### Zeta potential measurements

The zeta potentials of "bacteria-kaolinite" mixture colloids as functions of the bacteria contents were measured in DI water using ZETA Plus (Brookhaven Instruments Corporation, New York, USA) at 25 °C, pH 8.0. The samples for the zeta potential measurements were prepared in a similar manner as those for the aggregation experiments. The total mass concentration of the sample suspension was 300 mg/L, and the mass concentrations of the bacteria in the mixed suspension were set as 0, 10, 20, 50, and 100 mg/L, respectively. Triplicate measurements were performed with ten runs per measurement.

## RESULTS AND DISCUSSION

### Kinetics of "kaolinite *-P. putida*" aggregation

The curves describing the hydrodynamic diameter growth of "kaolinite-*P. putida*" mixture colloids in $LiNO_3$ solutions were shown in Fig. 1. It could be seen that the
hydrodynamic diameters of "kaolinite-*P. putida*" aggregates increased with increasing $LiNO_3$ concentrations. For example, the hydrodynamic diameters of mixture colloids containing 100% kaolinite and 0% *P. putida* indicated that aggregation occurred between kaolinite and kaolinite. However, the hydrodynamic diameter of the mixture colloids containing 0% kaolinite and 100% *P. putida* suggested that aggregation would not occur between *P. putida* and *P. putida*. The mixture colloids aggregated when the proportion of *P. putida* was 0%, 3.33%, 6.67%, 16.67% and 33.33%, but the aggregation rate decreased with increasing proportion of *P. putida*.

The curves describing the hydrodynamic diameters of different "kaolinite-*P. putida*" mixture as a function of time in $KNO_3$ solutions were shown in Fig. 2. Similar to the aggregation in $LiNO_3$ solutions, the hydrodynamic diameters of "kaolinite-*P. putida*" mixture colloidal aggregates increased with increasing $KNO_3$ concentrations. Additionally, the aggregation can occur between kaolinite and kaolinite but not occur between *P. putida* and *P. putida*. The mixture colloids aggregated when the proportion of *P. putida* was 0%, 3.33%, 6.67%, 16.67% and 33.33%, but the aggregation rate decreased with increasing proportion of *P. putida*.

The curves describing the hydrodynamic diameter of different proportions of "kaolinite-*P. putida*" as a function of time in $CsNO_3$ were shown Fig. 3. Similar to colloidal aggregation in $LiNO_3$ and $KNO_3$ solutions, the hydrodynamic diameters of "kaolinite-*P. putida* " mixture colloidal aggregates increased with the increase of $CsNO_3$ concentration. The aggregation can occur between kaolinite and kaolinite but not between *P. putida* and *P. putida*.

## TAA rates and CCC

Using the experimental data given in Figs. 1, 2 and 3, the TAA rates $\tilde{v}_T(f_0)$ of mixture colloids aggregation in various alkali ion solutions were calculated according to Eq. (1), and their relationships with the alkali ion concentration $f_0$ were given in Table 1. The more detailed results of the TAA rates changing with electrolyte concentrations has been given in the Supporting Information (Figs. S1–S5).

As can be seen, the CCC of the mixture colloids under different bacterial contents can be obtained (Table 1). For any given electrolyte solutions, the CCC for the mixture colloids increased as the bacterial content increased. For the aggregation of the mixture colloids with the same bacterial contents, the CCC increased in the order of $Li^+ > K^+ > Cs^+$, exhibiting strong specific ion effects. For example, when the *P. putida* content was 3.33%, the CCCs for the aggregation of mixture colloids were 18.7, 16.7 and 9.4 mmol/L for $Li^+$, $K^+$ and $Cs^+$, respectively. Moreover, the specific ion effects reflected by the CCC values increased as the bacterial content increased.

When the electrolyte concentrations were less than the CCC, the aggregation belongs to the RLCA regime, and the repulsive potential energy between the colloidal particles was higher than the attractive potential energy. Table 1 showed the fitting equation for aggregation of "kaolinite-*P. putida* " mixture colloids in different electrolyte solutions when the electrolyte concentrations were less than the CCC. As can be seen, given the same electrolyte conditions, the TAA rates of "kaolinite-*P. putida*" mixture colloids decreased

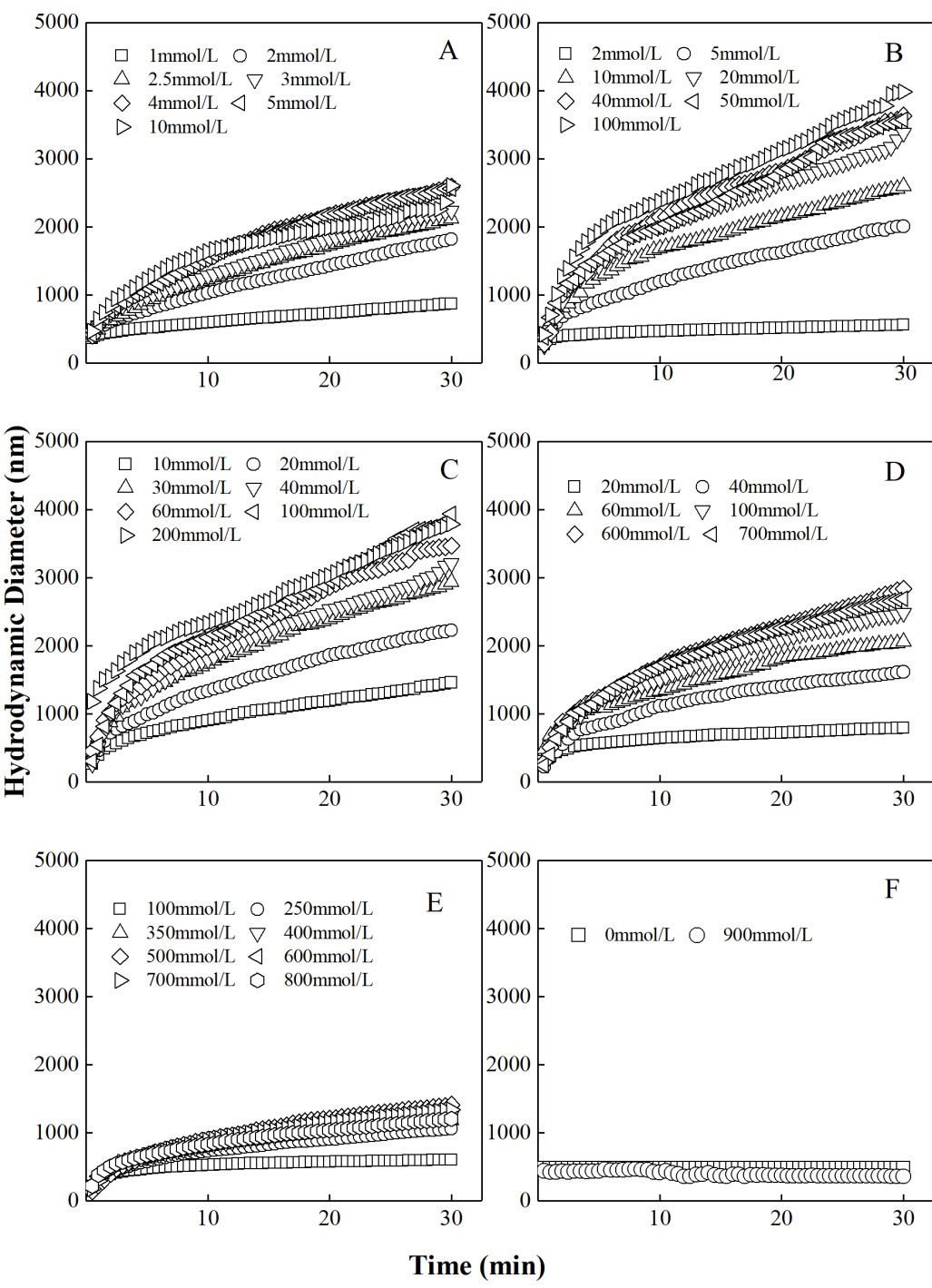

**Figure 1** Growth of hydrodynamic diameters of "100%kaolinite (A), 96.67% kaolinite + 3.33% *P. putida* (B), 93.33% kaolinite + 6.67% *P. putida* (C), 83.33% kaolinite + 16.67% *P. putida* (D), 66.67% kaolinite + 33.33% *P. putida* (E), 100% *P. putida* (F)" aggregates in $LiNO_3$ solutions.

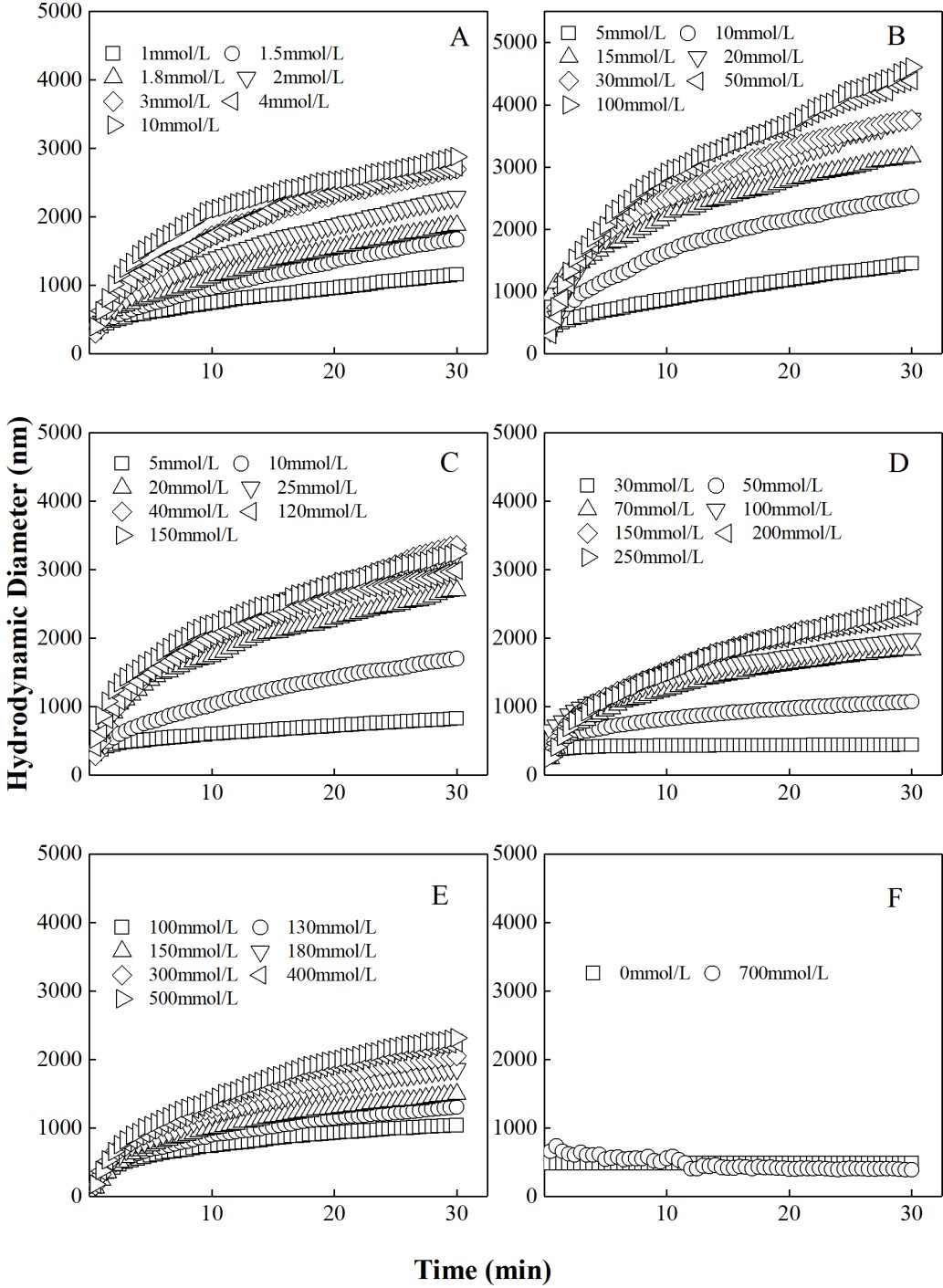

**Figure 2** Growth of hydrodynamic diameters of "100%kaolinite (A), 96.67% kaolinite + 3.33% *P. putida* (B), 93.33% kaolinite + 6.67% *P. putida* (C), 83.33% kaolinite + 16.67% *P. putida* (D), 66.67% kaolinite + 33.33% *P. putida* (E), 100% *P. putida* (F)" aggregates in $KNO_3$ solutions.

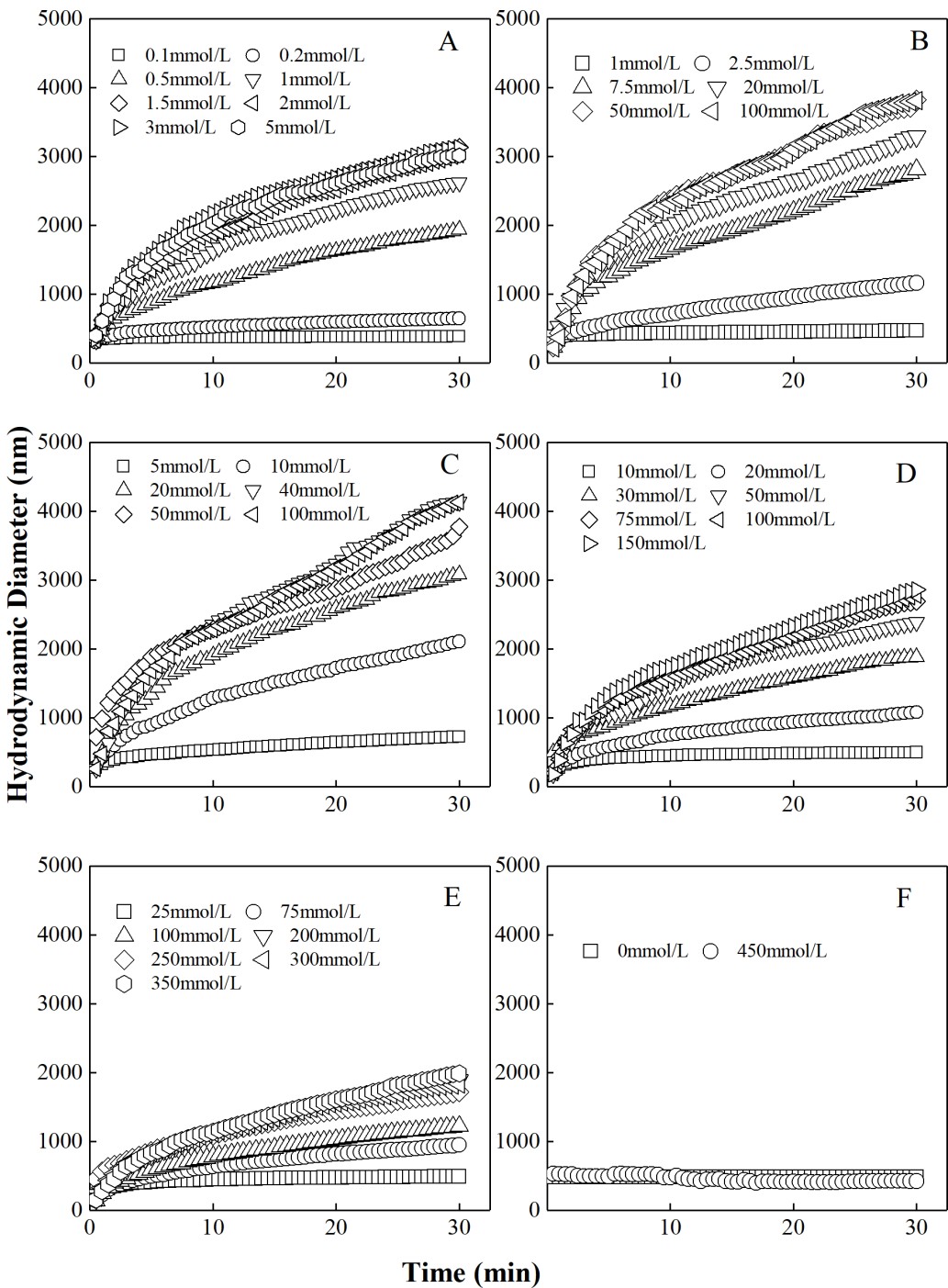

**Figure 3** Growth of hydrodynamic diameters of "100%kaolinite (A), 96.67% kaolinite + 3.33% *P. putida* (B), 93.33% kaolinite +6.67% *P. putida* (C), 83.33% kaolinite + 16.67% *P. putida* (D), 66.67% kaolinite + 33.33% *P. putida* (E), 100% *P. putida* (F)" aggregates in $CsNO_3$ solutions.

**Table 1** Expressions of the TAA rates for the aggregation of mixed particles in the various alkali ion solutions when the electrolyte concentration is less than the CCC.

| P. putida contents (%) | CCC values (mmol/L) | | | Expressions of TAA rates vs. electrolyte concentration | | |
|---|---|---|---|---|---|---|
| | Li$^+$ | K$^+$ | Cs$^+$ | Li$^+$ | K$^+$ | Cs$^+$ |
| 0 | 3.2 | 2.8 | 0.89 | $40.05f_0$-18.52 | $54.64f_0$-23.68 | $195.09f_0$-16.64 |
| 3.33 | 18.7 | 16.7 | 9.4 | $8.77f_0$-8.92 | $13.02f_0$-16.92 | $18.24f_0$-10.66 |
| 6.67 | 38.4 | 30.3 | 26.2 | $4.11f_0$-9.56 | $5.65f_0$-13.07 | $8.22f_0$-14.67 |
| 16.67 | 71.2 | 61.4 | 51.3 | $1.67f_0$-6.12 | $1.90f_0$-14.25 | $2.40f_0$-15.05 |
| 33.33 | 372.2 | 187.9 | 182.6 | $0.12f_0$-4.697 | $0.54f_0$-18.23 | $0.40f_0$-5.01 |

as the bacterial content increased. On the other hand, for the same bacterial contents, the TAA rates for the aggregation of "kaolinite-*P. putida*" mixture colloids increased in the order of Li$^+$ < K$^+$ < Cs$^+$, exhibiting strong specific ion effects. For example, when the *P. putida* content was 3.33%, the TAA rates for the aggregation of mixture colloids were 1.21, 48.2, and 80.5 nm/min for Li$^+$, K$^+$ and Cs$^+$, respectively.

## Interaction between *P. putida* and kaolinite and the aggregation mechanism

The activation energies for the aggregation of the mixture colloids in the various alkali ion solutions were calculated using Eq. (2) and then plotted in Fig. 4. At any given same electrolyte concentrations below CCC, the activation energies for the mixture colloids with different bacteria contents were significantly different. In addition, the activation energies increased as the bacterial content increased. At any given different alkali ion concentrations below CCC, the activation energies for the mixture colloids with the same bacteria contents increased in the order of Li$^+$ > K$^+$ > Cs$^+$. For example, when the *P. putida* content was 3.33% and the electrolyte concentration was 5 mmol/L, the activation energy for the aggregation of mixture colloids were respectively 1.5$RT$, 1.4$RT$, and 0.7$RT$ for Li$^+$, K$^+$ and Cs$^+$.

The activation energies for the mixture colloids aggregation might come from the electrostatic repulsion potential as well as the molecular attraction potential between *P. putida* and kaolinite. The electrostatic repulsion potential could be characterized by the zeta potential, we therefore determined the zeta potential of "kaolinite-*P. putida*" mixture colloids, and the results were shown in Fig. 5. The Zeta potential of kaolinite at pH = 8 is −44 mv, which is not far from the −46 mv reported by *Feng et al. (2020)* at pH = 7. However, The Zeta potential of *P. putida* at pH = 8 is −37 mv, which is far from the −8.5 mv reported by *Shamim & Rehman (2014)* at pH = 6.2. We suspect that it may be related to pH and the pretreatment process of bacteria.

From Fig. 5, we could see that the electrostatic repulsion potential between *P. putida* and kaolinite decreased as the *P. putida* content increased. In other words, the activation energy for the mixture collolds aggregation should decrease as the *P. putida* content increased. According to the experimental results shown in Fig. 4, the activation energy for the mixture collolds aggregation, however, increased with the increase of *P. putida*

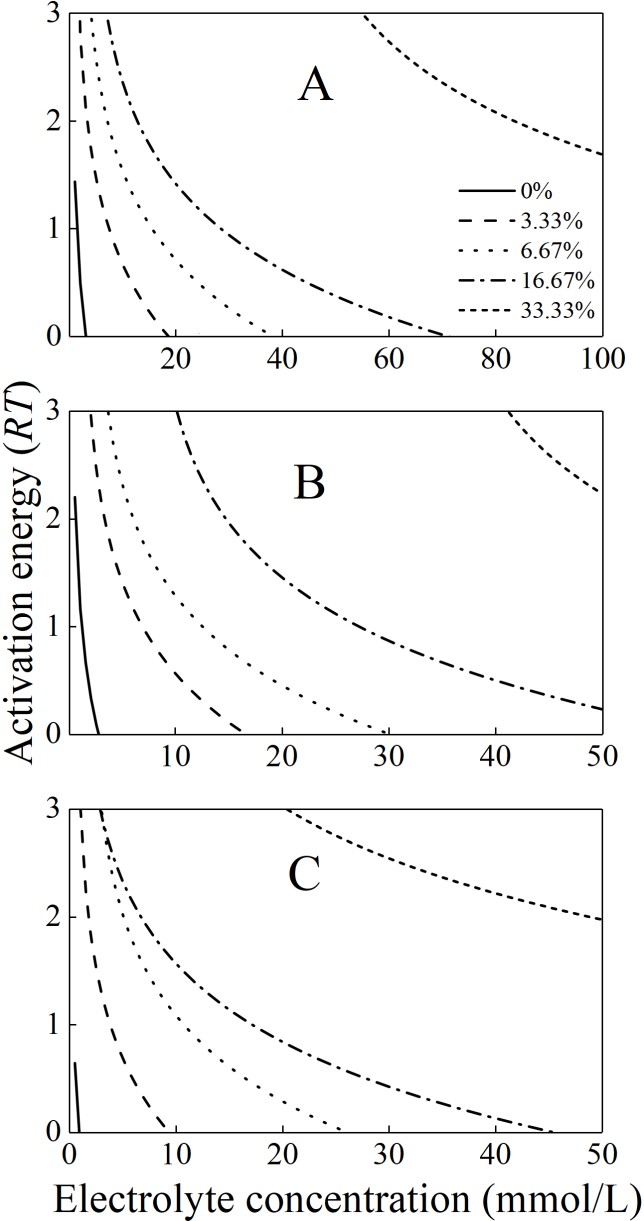

**Figure 4** The activation energies for the aggregation of the mixture colloids with different bacteria contents in (A) LiNO$_3$ (B) KNO$_3$, and (C) CsNO$_3$ solutions.

content. Therefore, the only possible explanation for the decrease in the aggregation rate with increasing bacterial content could be that the Hamaker constant of the bacteria is significantly lower than that of kaolinite. In this case, the apparent Hamaker constant of the mixture colloids decreased with increasing *P. putida* content, which would further lead to the decrease of molecular attractive energies between colloidal particles.

However, apart from the aggregation between *P. putida* and kaolinite in the mixture colloids, another two kinds of aggregation process might occur: the aggregation between

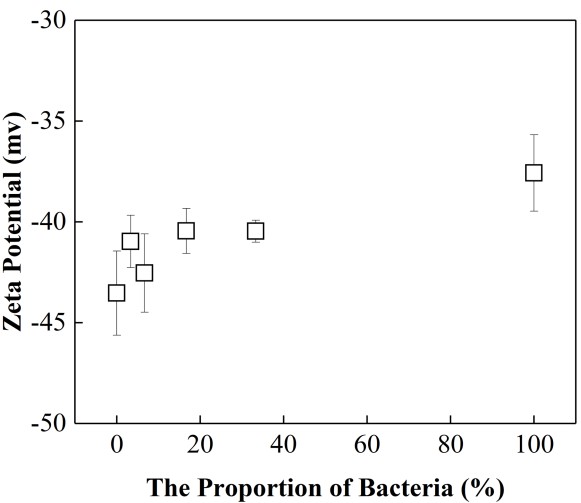

**Figure 5** Zeta potential of "kaolinite-*P. putida*" mixed particle as a function of *P. putida* content at pH 8.0.

*P. putida* and *P. putida* as well as the aggregation between kaolinite and kaolinite. We can speculate that there may be three CCCs in the mixture colloids: the CCC of kaolinite-kaolinite aggregation, the CCC of *P. putida*-*P. putida* aggregation and the CCC of kaolinite-*P. putida* aggregation. According to the experimental results, we can find that, the CCC for pure kaolinite colloids (kaolinite-kaolinite aggregation) is the lowest; the CCC for 100% *P. putida* colloids (*P. putida*-*P. putida* aggregation) is infinite. Therefore, we can speculate that the CCC for kaolinite-*P. putida* aggregation would be in the middle. Therefore, based on this analysis, at a given *P. putida* content and a given cation type, the variation of TAA rates changing with electrolyte concentrations for the mixture colloids should exhibit the characteristics as shown in Fig. 6. The first straight line at lower electrolyte concentrations should represent the aggregation between kaolinite and kaolinite because the activation energy for the aggregation between kaolinite and kaolinite is the lowest. The activation energy for the mixture colloids greatly reduced when the electrolyte concentration gradually increased higher than the first turning point. The aggregation of *P. putida* and kaolinite became important. Therefore, the second straight line contained the rapid aggregation of kaolinite and kaolinite and the slow aggregation of *P. putida* and kaolinite. The third straight line represented the rapid aggregation of *P. putida* and kaolinite. Therefore, the electrolyte concentration corresponding to the second inflection point is the CCC of *P. putida* and kaolinite.

However, in the mixture colloids, just one CCC exhibited for given mixture suspension, and for different mixture suspensions, the CCC increased as the content of *P. putida* increased (Table 1). Table 1 showed that, (1) the two turning points shown in Fig. 6 could not be observed from the experimental results of the *P. putida* and kaolinite mixture colloids. (2) The CCC of kaolinite and kaolinite was not observed from the mixture colloids. (3) There is only one CCC obtained for the mixture colloids. Based on this, we can deduce that the aggregation behavior of the mixture colloids is determined by the average

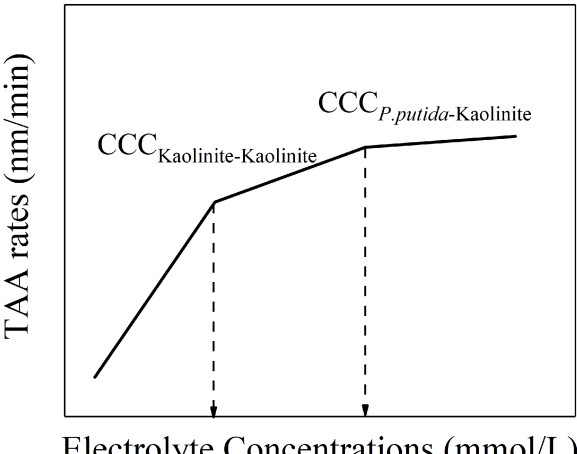

**Figure 6  Schematic diagram of "kaolinite-*P. putida*" TAA rates as a function of electrolyte concentration.**

effects produced by the various colloidal particles instead of separate colloidal particle. The experimental results shown in Table 1 could be satisfactorily explained by this inference. Each mixture colloidal system can only have one specific CCC, which is determined by each physical parameter (such as determination of the average value of potential, electric field strength, charge density, Hamaker constant, etc.) since each physical parameter of the mixed system (such as potential, electric field strength, charge density, Hamaker constant, etc.) has only one average value. What's more, the average Hamaker constant of the mixture colloids decreased with the increase of *P. putida* content due to higher Hamker constant of kaolinite and lower Hamaker constant of *P. putida*. Eventually, the CCC of mixed system increases with the increase of *P. putida* content.

Finally, we would like to give a speculated explanation for the experimental results for the kinetics of pure *P. putida* aggregation. We know that, *P. putida* is a gram-negative, rod-shaped bacteria, and with the polar flagellum or with positively and negatively charges at cell surfaces; in addition, generally, bacteria prefer to survive in cell groups but not a single cell. Therefore, *P. putida* could aggregate easily. In our experiments, however, (1) the net charge of *P. putida* was negative, therefore, the electrostatic fore between two cells must be repulsive as the distance between two cells was relative long; (2) the adopted cations in this study were monovalent cation, those cations just could produce relative weak screening effect on the electric field around cell, thus a long range electrostatic repulsive force between two cells could present; (3) the density of cell particles in this experiments was very low, thus the long range electrostatic repulsive force dominate cells interaction. Those three points might be the reasons for explaining the kinetics of pure *P. putida* aggregation. Therefore, on the contrary, if the cell density was very high, and if divalent cation presented in the suspension, the cell aggregation might be possibly observed.

## CONCLUSIONS

In this paper, the aggregation behaviors of *P. putida*, kaolinite and "kaolinite-*P. putida*" mixed colloids in various concentrations of $LiNO_3$, $KNO_3$ and $CsNO_3$ electrolytes were studied. Aggregation occured between kaolinite and kaolinite whereas no aggregation occured between *P. putida* and *P. putida* in the presence of any kind of the three electrolytes. Additionally, the aggregation occured in the mixture colloids when the bacterial content in the mixed system was less than 50%. TAA rate decreased with the increasing bacterial content. Additionally, specific ion effects affected the aggregation of "*P. putida* -kaolinite" mixture colloids. The TAA rates for the aggregation of the mixture colloids with same bacteria content increased in the order of $Cs^+ > K^+ > Li^+$. For example, when the *P. putida* content equaled 3.33%, the TAA rates for the aggregation of the mixture colloids were 1.21, 48.2, 80.5 nm/min for $Li^+$, $K^+$ and $Cs^+$ respectively. The CCC for the aggregation of the mixture colloids with the same bacteria content increased in the order of $Li^+ > K^+ > Cs^+$. The CCC for the aggregation of the mixture colloids were 18.7, 16.7 and 9.4 mmol/L for $Li^+$, $K^+$ and $Cs^+$, respectively, when the *P. putida* content equaled 3.33%. The aggregation activation energy for the aggregation of the mixture colloids with the same bacteria contents increased in the order $Li^+ > K^+ > Cs^+$. The increasing of aggregation activation energy and decreasing of TAA rates for the aggregation of the mixture colloids as *P. putida* content increased might come from the lower Hamaker constant of *P. putida* than that of kaolinite, which further gave rise to lower molecular gravitational potential between colloidal particles. What's more, the aggregation behavior of a mixed colloidal system was determined by the average effect of electrostatic potential and molecular gravitational potential produced by the various colloidal particles instead of separate colloidal particle. This finding provides an important methodological guide for studying the aggregation behavior of "bacterial-clay" mixed systems.

### Funding

This work was supported by the National Natural Science Foundation of China (grant numbers 41501241 and 41530855); Fundamental Research Funds for the Central Colleges (grant number XDJK2019B037). The funders had no role in study design, data collection and analysis, decision to publish, or preparation of the manuscript.

### Grant Disclosures

The following grant information was disclosed by the authors:
National Natural Science Foundation of China: 41501241, 41530855.
Fundamental Research Funds for the Central Colleges: XDJK2019B037.

### Competing Interests

The authors declare there are no competing interests.

## Author Contributions

- Zhaoxuan Yu conceived and designed the experiments, performed the experiments, analyzed the data, performed the computation work, prepared figures and/or tables, authored or reviewed drafts of the paper, and approved the final draft.
- Rui Tian conceived and designed the experiments, analyzed the data, prepared figures and/or tables, authored or reviewed drafts of the paper, and approved the final draft.
- Dian Liu and Yekun Zhang performed the experiments, analyzed the data, and approved the final draft.
- Hang Li conceived and designed the experiments, analyzed the data, performed the computation work, prepared figures and/or tables, authored or reviewed drafts of the paper, and approved the final draft.

## Data Availability

The raw measurements are available in the Supplemental Files.

## Supplemental Information

Supplemental information for this article can be found online at http://dx.doi.org/10.7717/peerj-pchem.12#supplemental-information.

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
