# Peer review of "Aggregation kinetics of binary systems containing kaolinite and *Pseudomonas putida* induced by different 1:1 electrolytes: specific ion effects"

_PeerJ Physical Chemistry, doi:10.7717/peerj-pchem.12_

## Round 0.1 · original submission · Major Revisions

Please pay particular attention to the concerns of Reviewer 3 when revising your manuscript.

In addition to addressing the scientific issues raised by the reviewers, I recommend improving the English usage in the manuscript, especially article use and subject-verb agreement.

Please also note that you are not obliged to cite any of the references suggested by the reviewers (i.e., point 9 from Reviewer 1 and points b and c from Reviewer 4). Please only cite these works if they are highly relevant to the present submission.

Reviewer 1 ·

Basic reporting

The paper appeared with interesting results and new messages are given. The paper is of a satisfactory level, quite clear and well organized.

Experimental design

All the methods used are well established and the discussion part is complete.

Validity of the findings

The contribution of this piece of research is valuable

Additional comments

The aim of this work was to study the aggregation of "mineral-bacteria" mixture colloids of “kaolinite-P. putida” induced by LiNO3, KNO3 and CsNO3. Dynamic light scattering studies on the aggregation kinetics of mixture colloids containing kaolinite and P. putida were conducted in this study. The experimental study is well organized, providing new insides in the field of colloidal aggregation behavior of binary systems.
Minor comments
1. The XDLVO theory should be used to explain the aggregation between kaolinite-kaolinite, kaolinite-P. putida, and P. putida-P. putida.

2. L123: The hydrodynamic diameter of P. putida was measured as 1600±100 nm. How did you measure the diameter?
3. L143: ….indicating the P. putida contents were 0%, 144 3.33%, 6.67%, 16.67%, and 33.33%. Based on weight? Please explain.

4. Eqs. 1 and 2: vT(f0) does not appear in the equation 1. Present clearly the vo/to(f0).

5. The procedure for the Kinetics of “kaolinite-P. putida” aggregation, should be described in the methods section.

6. Please correct the typo mistakes throughout the text. E.g. scattering, Oncsik, et al(Oncsik et al. 2015).

7. L261: We can speculate that there may …….. three different conditions since different electrostatic repulsion … kaolinite-kaolinite, kaolinite-P. putida, and P. putida-P. putida. Please use literature findings to support the statements appeared in the text.

8. L282: Based on this, we can deduce that the aggregation behaviour…….average effect produced by the various colloidal …….of separate colloidal particle. Please use literature findings to support the statements appeared in the text.

9. Please consider also the publications in this field: doi:10.1029/2010WR009560, doi:10.1016/j.colsurfb.2011.01.026

·

Basic reporting

Clear English;
Good literature;
Professional article structure, figures, tables. Raw data shared;

Experimental design

Well enxperimental design

Validity of the findings

This paper shows the novelty;
All underlying data are robust, statistically sound, & controlled;
Conclusions are well stated;

Additional comments

This study showed dynamic light scattering studies on the aggregation kinetics of mixture colloids containing kaolinite and Pseudomonas putida(P.putida) . This is a very good topic to study. The results showed that the aggregation behavior of mixture colloids or binary systems was determined by the average effects, rather than the specific component, of binary systems. The experiment design is good and English is good. I suggest that this paper can be accepted.

Reviewer 3 ·

Basic reporting

In this paper, the authors investigated the colloidal interactions between P. putida and kaolinite in the different mass ration by dynamic light scattering. The results showed that aggregation happened in kaolinite colloidal particles, and between kaolinite and bacterial cells, but not happened in bacterial cells. This study provides interesting data for understanding the aggregation phenomena in soil and sediment. However, several aspects should be improved before publication.

Experimental design

This paper is original research within the Aims and Scope of the journal.

Validity of the findings

Speculation is welcome, but should be identified as such.

Additional comments

Firstly, all the aggregation data were obtained by dynamic light scattering in 30 min. The authors don't tell us why 30 min, and is it enough for aggregation experiment? And what would happen if conducted a more prolonged test, for example, 60 min or 120 min?
Secondly, P. putida is a gram-negative, rod-shaped bacteria, and with the polar flagellum. Generally, bacteria prefer to survive in cell groups but not a single cell. Furthermore, plenty of positively and negatively charged surface sites existed on the cell surfaces. Therefore, it is hard to understand why the bacteria could not aggregate in the present study. The authors need to provide more explanation on this point.
Thirdly, the aggregation between mineral colloid and bacteria could be the first step in forming soil aggregates. It is essential but not enough to explain the total aggregation phenomena in soil. More aggregation work is needed to conduct, e.g. at different pHs, to explain the aggregation better.

Reviewer 4 ·

Basic reporting

no comment

Experimental design

The experimental design is thoroughly described

Validity of the findings

The conclusions are very well stated

Additional comments

peerj (#47222): Aggregation kinetics of binary systems containing kaolinite and Pseudomonas putida induced by 1:1electrolyte.

This study examines the interactions between kaolinite colloidal particles and P. putida. Certainly, the research topic is of great interest because the migration of a mixture of clay particles and bacteria in environmental systems is not fully understood. The manuscript is very well written and organized. The experimental procedures are thoroughly presented. The figures are clear. Consequently, this reviewer recommends publication of this manuscript in PeerJ after a relatively minor revision. The following is a short list of minor suggestions:

(a) The abstract should clearly list all the novel contributions of this work.

(b) The introduction section of the manuscript is very informative; however, the interaction of kaolinite colloidal particles with various biocolloids as well as numerous contaminants has been studied extensively in the literature. Therefore, the introduction should be expanded to mention previous studies such as the works by Chrysikopoulos and Syngouna (Colloids and Surfaces B: Biointerfaces, 92, 74–83, 2012), Bellou et al (Science of the Total Environment, 517, 86–95, 2015) and Fountouli et al (Environmental Earth Sciences, 78, 152, 2019), Fountouli et al (Environmental Earth Sciences, 78, 152, doi:10.1007/s12665-019-8147-x, 2019).

(c) Note that Vasiliadou et al (Colloids and Surfaces B: Biointerfaces, 84(2), 354–359, 2011) have studied the interactions between different structured kaolinite materials and P. putida. Also Vasiliadou and Chrysikopoulos (Water Resources Research, 47(2), W02543, doi:10.1029/2010WR009560, 2011) investigated the interactions between kaolinite and P. putida during transport in porous media.

(d) How do the measured zeta potentials compare with literature values?

(e) The captions could be more descriptive. The captions for Figures 1-3 need more attention.

(f) Can Figure 6 contain some quantitative information? Currently Figure 6 is not very informative.

---

## Round 0.2 · accepted · Accept

I am satisfied that all of the scientific issues raised by the reviewers have been adequately addressed in the revised manuscript. However, the manuscript may still require editing to improve English usage. Please liaise with PeerJ staff about the required improvements (if any).

Reviewer 3 ·

Basic reporting

No comment

Experimental design

No comment

Validity of the findings

No comment

Additional comments

All of my concerns from my previous review have been addressed. I recommend an accept with some improvement of writing and language polishing.